# Mood Status Response to Physical Activity and Its Influence on Performance: Are Chronotype and Exercise Timing Affect?

**DOI:** 10.3390/ijerph20042822

**Published:** 2023-02-05

**Authors:** Hengxu Liu, Jiaqi Liang, Kun Wang, Tingran Zhang, Shiqi Liu, Jiong Luo

**Affiliations:** Research Centre for Exercise Detoxification, College of Physical Education, Southwest University, Chongqing 400715, China

**Keywords:** mood status, physical activity, exercise timing, circadian rhythm, chronotype, systematic review

## Abstract

Purpose: It is well known that there is an obvious 24 h diurnal variation in the individual’s mood state and physiological activity, and training at different times of the day may lead to different exercise performance and metabolic outcomes; however, the time-dependent effect of emotional state on physical activity and the influence of its circadian rhythm on exercise performance are still not comprehensively understood. Based on this, this study summarizes the rhythmic experimental research in the field of sport psychology, and it aims to provide the basis for coaches to optimize sports training scientifically and to improve the mental health of the related crowd to the greatest extent. Methods: The systematic review was performed in accordance with the Preferred Reporting Items for Systematic Reviews and Meta-Analyses guidelines. We searched the PubMed, Web of Science, Medline, and CNKI databases for relevant literature; the search scope was research before September 2022. Results: 13 studies comprising 382 subjects examined the effects of exercise timing on mood responses to exercise or the effects of circadian rhythms of mood on exercise performance, which included 3 RCTs and 10 Non-RCTs. The subjects included athletes (both training or retired), college students, and healthy adults. Two studies were designed for long-term exercise intervention (aerobic training and RISE) and the rest for acute intervention (CrossFit training, HIIT, aerobic combined with muscle conditioning training, constant power exhaustion training, and cycling) or physical function tests (RSA + BTV tests, 30 s Wingate test, muscle strength + CMJ + swimming performance test, RSSJA, shooting accuracy tests + 10 × 20 m dribbling sprint, 200 m time trials). All trials reported specific exercise timing; of these, 10 studies reported subjects’ chronotypes, most commonly using the MEQ scale, while 1 recorded with the CSM. Mood responses were assessed with the POMS scale in 10 studies, while 3 other studies used the UMACL, PANAS, and GAS scales, respectively. Conclusion: There was much inconsistency between the results, with subjects likely to be exposed to more sunlight (the main timing factor of the circadian rhythm) during early morning exercise, resulting in feeling more positive emotions; however, following a night’s rest, delayed responses and poor functioning of the various organ systems of the human body may also lead to higher feelings of fatigue and negative emotions indirectly. Conversely, for athletes, their physical function tests are also more susceptible to the circadian rhythm of emotions, suggesting the importance of synchronizing them. In addition, night owls’ emotional state during physical activity seems to be more susceptible to exercise timing than that of early birds. In order to achieve the best emotional state, it is suggested that night owls arrange courses in the afternoon or evening in future training.

## 1. Introduction

The circadian rhythm of mammals is a 24 h physiological and behavioral cycle produced by the internal oscillations of the biological clock [1,2]. Its suprachiasmatic nucleus (SCN, central clock) can sense external natural light and send the carried timing signal to the peripheral biological clock in the lung, intestine, etc. [3,4]; the peripheral circadian rhythm is not only regulated by SCN directly but also pulled by the local microenvironment, which leads to the tissue specificity of biological clocks [5]. It is driven by this mechanism that results in obvious circadian alternation in gene expression at the molecular level as well as in the sleep–wake cycle, motor performance, psychological cognition, etc., in exogenous behaviors [6,7], and it is especially important to arrange work content at the proper time of day based on an individual’s physiological and psychological rhythms.

Chronotype refers to an individual’s natural inclination to circadian rhythms, which can be divided into three types by daily activity and sleep time: morning type (M-type), evening type (E-type), and neither type (N-type) [8]. M-type tends to go to bed early and schedule more physical activity in the morning, E-type, commonly known as “night owls”, prefers to go to sleep and wake up late and show lower subjective and objective sleep quality [9], while N-type schedules come in between [10]. The Morningness–Eveningness Questionnaire (MEQ) and Munich Chronotype Questionnaire (MCTQ) are the most commonly used measurements of chronotype. Chronotype reflects a person’s biorhythm and behavioral phenotype. Individuals with different chronotypes vary in hormone secretion, sport performance, jet lag, self-efficacy, etc. [11]. In addition, it is also associated with psychological functioning and mood states; for instance, a positive correlation between E-type and depressive symptoms in adolescents has been demonstrated [12,13]. Moreover, the E-type is prone to show higher levels of subjective fatigue in work tasks, which may be related to the shortened duration of sleep [14]. The M-type tends to face dilemmas with a positive attitude, thus alleviating the subjective experience of negative emotions [15], all suggesting that chronotype may act as a potential predictor of personality and emotional state in daily activities.

Human exercise is a highly unified composite performance of physiological and psychological activities, as variables such as core body temperature, hormone secretion, muscle contractile strength, and VO_2_ max, which are closely related to exercise response and adaptation, all show obvious diurnal fluctuations, leading to different physical performance and metabolic outcomes between morning and afternoon/evening exercise [16,17,18,19,20]. In recent years, whether the response of mood state to physical activity (during and after exercise) is time-dependent has also attracted the attention of scholars. As we all know, physical exercise could administer to reduce depression and anxiety, and increase energy arousal and happiness [21,22,23]. Conversely, the appropriate emotional state contributes to reducing the inertia of training and regulates the body’s physiological response by a series of molecular signaling pathways in the brain, indicating the important dynamic interaction between them. Due to the diurnal fluctuation of an individual’s emotion and the rhythmic change of the exercise system [24,25], it can be inferred that there exists an optimal exercise opportunity in a day to reach the most appropriate emotional state, thus improving compliance to training and amplifying the health-promoting benefits of exercise through the nervous, endocrine, and immune systems (both physical and psychological), and the diurnal variation in mood status (before training) could also exert an influence on exercise performance. In addition, the chronotype of individuals is also a factor that cannot be ignored. Distinct day–night patterns mean different bedtimes and the time between exercise and wake-up, which may indirectly affect the mental state during exercise (such as fatigue and vigor). However, it remains unclear how the time of day and chronotype regulates mood states’ response to physical activity and how the circadian rhythm of mood affects performance. Based on this, the systematic review summarized the evidence on rhythmicity in sports psychology. The innovation of this study is that it is the first time to review the interaction between chronobiology and exercise psychology in different situations, and it aims to provide a reference for relevant people to maximize the emotional benefits from exercise and for coaches to arrange training more scientifically.

## 2. Method

The systematic review was performed in accordance with the Preferred Reporting Items for Systematic Reviews and Meta-Analyses guidelines [26].

### 2.1. Literature Retrieval Strategy

Two reviewers (L.H.X. and L.J.Q.) searched PubMed, Web of Science, Medline, and CNKI databases for relevant literature on 2 November 2022. The search scope was all studies before September 2022. The retrieval strategy was based on the combination of subject words and free words, and using the Boolean operators “AND” and “OR” to connect. Taking Web of Science as an example, the specific retrieval strategy was as follows.

#1 Mood state OR Psychological response OR Mental state OR Emotional state;#2 Physical activity OR Exercise OR Training OR Sports OR Sports performance OR Exercise performance;#3 Chronotype OR Exercise timing OR Time of day OR Circadian rhythm OR Diurnal variation OR Diurnal rhythms;#4 #1 AND #2 AND #3.

In addition, we screened the reference lists from the included literature for potential studies that had not been previously retrieved.

### 2.2. Inclusion and Exclusion Criteria

The inclusion criteria included: (1) The intervention study of the time-dependent effects of emotional states during or following exercise or reporting the effects of chronotype on mood responses to exercise; (2) The study on how the circadian rhythm of mood affects exercise performance; (3) The type of research is self or randomized controlled trial; (4) The subjects had no congenital mental disorders.

The exclusion criteria included: (1) republished studies; (2) not written in English; (3) without peer review; (4) animal experiments, retrospective studies, case reports, meeting abstracts, and guidelines.

### 2.3. Literature Information Extractions

Using Endnote x9 for document screening and management, two reviewers (L.J.Q. and L.S.Q., they performed screening exercises in advance) independently screened the literature, extracted data (using a data extraction form based on the ICF framework) and cross-checked it, and left it to a third party for discussion if there existed any dispute. Information extracted included the type, author, year, country, and subject’s basic characteristics of the study, as well as the mode, cycle, intensity, frequency of the exercise intervention, and outcome measures. After the first screening, the consistency among screeners reached 92.3%, and came to 100% after the third party’s discussion on the dispute.

### 2.4. Bias Risk Assessments

The Cochrane randomized controlled trial (RCT) risk of bias tool [27] was independently used by two reviewers (L.H.X. and L.J.Q.) to assess the risk of bias in the randomized controlled trials; if the opinion could not reach a consensus, it would be referred to a third party to decide. The evaluation criteria of Cochrane mainly include random sequence generation, allocation concealment, blinding of researchers and subjects, blinding of outcome testing, complete outcome data, and other biases. In each dimension, literature risk of each dimension is rated as “low risk”, “high risk” or “unclear risk”.

The bias risk of the non-randomized controlled trial (Non-RCT) studies was evaluated with the ROBIN-S scale [28] by two reviewers (L.H.X. and L.J.Q.), which included confounding bias, selection bias, bias in measurement classification of interventions, bias due to deviations from intended interventions, bias due to missing data, bias in the measurement of outcomes, and bias in the selection of the reported result, and the result was assigned as “low risk”, “moderate risk” or “serious risk” and “critical risk”. Likewise, if the two reviewers disagreed on the assessment, the decision was left to a third party.

## 3. Results

### 3.1. Literature Search

The retrieval process and results are presented in Figure 1. A total of 1514 articles were retrieved, and 811 were left following 703 excluded duplicates. Through scanning the titles, abstracts, and keywords of the literature, 59 potential articles were obtained. The two reviewers then analyzed the full text and excluded 46 articles for the following reasons: unavailable data (*n* = 2); not corresponding topics (*n* = 11); not relevant outcomes (*n* = 26); wrong intervention (*n* = 2); and other reasons (*n* = 5). Finally, a total of 13 studies were included in the systematic review.

### 3.2. Study Characteristics

A total of 11 studies comprising 352 subjects examined the effects of exercise timing on mood responses to exercise, or the effects of circadian rhythms of mood on exercise performance, which included 3 RCTs and 10 Non-RCTs. The subjects included athletes (both training or retired), college students, and healthy adults. Two studies were designed for long-term exercise intervention (aerobic exercise and resistance, interval, stretching, endurance (RISE) training), and the rest for acute intervention (CrossFit training, high intensity interval training (HIIT), constant power exhaustion training and cycling) or physical function tests (repeated sprint ability (RSA) + ball-throwing velocity (BTV) tests, 30 s Wingate test, muscle strength + countermovement jump (CMJ) + swimming performance test, repeated shuttle sprint and jump ability (RSSJA), shooting accuracy tests + 10 × 20 m dribbling sprint and 200 m time trials). All trials reported specific exercise timing; of these, nine studies reported subjects’ chronotypes, most commonly using the MEQ scale, while Sławińska recorded with the composite scale of morningness (CSM). Mood responses were assessed with the profile of the mood states (POMS) scale in nine studies, while two other studies used the UWIST mood adjective checklist (UMACL) and positive and negative affect schedule (PANAS) scales, respectively.

### 3.3. Risk of Literature Bias

The Robins-1 scale was used to assess the bias risk of the included Non-RCTs (Table 1). The selection bias risk, bias in measurement classification of interventions, bias due to deviations from intended interventions, bias due to missing data, and bias in the selection of the reported result were all given low risks in the included studies. Four studies were at moderate risk of confounding bias (e.g., the subjects knew the purpose of the trial in advance, etc.), and six were assigned a high risk of bias in the measurement of outcomes.

For the three included RCTs, all studies showed a high risk of performance bias and unclear selection bias (Figure 2). Since the mood status was measured in the form of scales and the objectivity of other measures, the detection bias was assigned a low risk even without blinding the measurer. The risk of random sequence generation bias in one study is unclear. One had other risks of bias, and the rest were rated as low risk.

## 4. Outcomes

Four studies reported the influence of diurnal variations in subjects’ mood status on exercise performance [29,31,32,33] (Table 2). Essid et al. [29] performed RSA and BTA tests of handball players in the morning, noon, and evening, respectively, and found that the subjects showed markedly higher levels of anxiety, confusion, depression, anger, and fatigue before the ME (*p* < 0.001), and a more pronounced vigor before the AE and EE (*p* < 0.001), while the RSA and BTA tests were performed better in the afternoon and evening than in the morning (*p* < 0.001). Chtourou et al. [32] found that elite male judokas showed greater levels of vigor in the afternoon compared to the morning; in addition, the subjects’ jump height in CMJ3 and CMJ4 was noted better in the afternoon. For swimmers, the vigor and anger levels were significantly higher before training in the ME group than in the EE group (*p* = 0.000, *p* = 0.012, respectively), and although diurnal variations in muscle strength were observed during the test, there were no significant differences in swimming performance between groups (*p* > 0.05) [20]. Another separate study [33] performed shooting accuracy tests and a 10×20 m dribbling sprint in male children soccer players and observed that the EE group showed significantly higher pressure and depression before training than the ME group (*p* < 0.05), while there were no significant differences in anger, anxiety, vigor, fatigue, and confusion between the groups (*p* > 0.05). In terms of athletic performance, the dribbling performance of the AE group and EE group was better than that of the EE group (*p* < 0.05), but the differences in exercise performance between the groups cannot be explained by diurnal variations in mood states.

In the other studies related to chronotype [9,20,30,39], Hill et al. [30] tested college students on Wingate at different times of the day and observed that anger, depression, fatigue, and total mood disturbance (prior to the tests) increased significantly in M-types during the day compared with E-types (*p* < 0.01), and M-types showed a drop in vigor during the day while E-types showed a significant increase in it and a greater reduction (prior to the tests) in total mood disturbance (*p* < 0.01). The mean and peak power of the Wingate test were significantly higher in the AE and the EE groups than in ME group (*p* < 0.01). Two trials conducted high-intensity training interventions on subjects. An RCT of CrossFit intervention for athletes reported considerable improvement in energetic arousal hedonic tone and relief of tension following exercise (*p* < 0.05), and the ME can raise the emotion of the E-type and N-type group to the level of M-type [39]. Another acute HIIT intervention for college students noted that compared to the M-type and the EE group, the E-type was more fatigued and showed less vigor during morning training (*p* < 0.05) [9]. Similarly, a test of athletic performance by swimmers showed that the ME group and the M-type were observed with significantly lower fatigue and higher vigor before ME (*p* < 0.05), while the subjects who were accustomed to training at night and the N-type showed lower fatigue before EE (*p* < 0.05), and the M-type worked better in ME (*p* = 0.036) [20].

## 5. Discussion

Physical activity could stimulate the release of neurotransmitters such as dopamine and (-)-noradrenaline, increase the functional connection between the amygdala and the orbitofrontal cortex, and enhance the execution function, thus reducing anxiety and depression, as well as increasing euphoria and sense of accomplishment [40,41,42]. The results of the studies included in this review are in line with previous findings [43,44,45] that either regular long-term exercise, short-term exercise or a physical function test exerted varying degrees of positive effects on the psychological responses, such as relieving confusion, depression, and other negative emotions.

### 5.1. Time Dependence Influence of Training on Emotional State

Several studies in the review examined the time-dependent effects of mood status responses to physical activity, but the results are contradictory. The study by Maraki et al. [34] found that ME was associated with higher levels of positive mood, which may be due to the longer sunlight exposure of the subjects in the early ME Group. It is known that intrinsically photosensitive retinal ganglion cells on the human retina can receive sunlight signals and convert them into electrical signals, which are then transmitted to the downstream perithalamic nucleus, thereby regulating mood changes [46,47]. It is generally considered that individuals are more likely to experience fatigue and negative emotions when not stimulated by enough sunlight, while the opposite could help to improve their mood status. Thus, more exposure to sunlight might contribute to the subjects’ higher energy arousal, vigor, and positivity during ME. In addition, it may also be related to the day–night fluctuations of negative emotions themselves [48,49]. As Maraki et al. [34] and Sławińska et al. [39] observed in their study, the differences in emotional perception were detected before training; therefore, the more pronounced effect of ME on negative mood observed in the trail may be since the subjects’ negative mood status and emotional disturbance in the morning were higher, while the relief effect of EE may be masked by the decreased circadian rhythm of negative emotion, but this needs to be verified further.

However, the study by Arciero et al. [37] reported a distinct result, which is that following a 12-week RISE intervention at different times of the day, a more pronounced higher fatigue level was observed in subjects in the ME group compared to the EE group; nevertheless, the authors did not elaborate on the findings. As we all know, compared with the evening, the “physiological inertia” is usually generated in the morning, resulting in more difficulty for the human organs to reach the best state quickly. In addition, following a night of rest, the muscle viscosity, core temperature, and metabolic performance are in a “trough” state [50,51], which all lead to more laborious and higher fatigue for subjects during ME, especially when engaged in a high-intensity acute training or test. Therefore, from the perspective of the 24 h circadian rhythm of human physiological function, it may lead to greater fatigue and lower vigor during exercise. Except for the natural rhythms of physiological function, the time-dependent changes of physiological responses to physical activity may also indirectly lead to different mood status during or following ME and AE/EE. For instance, melatonin is not only a key factor in synchronizing the body’s circadian rhythm [49,52] but is also closely related to fatigue, depression, mood disorders, and despair [53,54,55]. As an external timing factor, physical activity could promote melatonin secretion [56], and the timing of exercise during the day also exerts a different effect on melatonin rhythms (e.g., EE often leads to a delayed phase of melatonin [57]), which may further affect emotional states through a series of molecular signaling pathways. However, whether the phase shift in hormone secretion can modulate mood remains to be verified. Thus, it can be seen that there is still some controversy about whether the impact of acute exercise on emotional state is time-dependent.

Notably, no appreciable differences between groups were reported in another 12-week jogging exercise intervention [38], which suggested that the response of emotional state to exercise training may be transient, and the influence of circadian rhythm will gradually disappear when the training period is extended. In addition, it is worth noting that the subjects in Irandoust and other studies were depressed people, due to the fact that these patients had obvious characteristics of severe morning symptoms and mild night onset, so there is also a possibility that the improvement of ME on depression may be covered up by its own circadian rhythm rise, which may also lead to insignificant differences between groups. Furthermore, the subjects’ level of training, sex (as observed in the study by Arciero et al. [37]), and age as well as the heterogeneity of exercise intervention type and intensity may all lead to different results.

### 5.2. The Effect of Individual Chronotype on Psychological Reaction to Exercise

The individual’s chronotype will also exert an influence on the psychological response to physical activity. Sławińska et al. [39] performed a CrossFit intervention on the athletes and found that compared with the E-type and the N-type, the M-type had a significantly greater positive mood (hedonic tone and energetic arousal) before ME and reached the same level following training, which may be due to the “emotional ceiling effect” of the M-type in the morning, which prevented it from benefiting more from exercise. Two studies supported the importance of synchronizing diurnal chronotypes with exercise timing [20,31], that is, for athletes with the E-type, early morning training appears to be more conducive to their feeling of higher vigor and may contribute to their subsequent exercise performance; the relevant explanation may be that ME means shorter sleep duration and worse objective sleep quality for the E-type [58]. Furthermore, as seen in athletes, the peak phase of motor performance in the E-type was markedly delayed after awakening compared with the N-type, resulting in more difficulties for the organs of the physiological system to enter the state rapidly, and may lead to a lack of vitality and several negative emotions indirectly compared to the M-type, which may help to explain the reason that E-type was more fatigued and showed less vigor during morning training in the study by Sławińska et al. [39].

Interestingly, Vitale et al. [9] observed that the emotional state and RPE during evening training were not affected by the subjects’ chronotype, which suggested that the mood status of “night owls” seems to be more susceptible to exercise timing than that of “early birds”. This may be related to the different types of personality traits and personality; as we mentioned in the introduction, M-type is apt to perform work tasks or confront difficulties with a positive and optimistic attitude, and show fewer mood swings, while E-type tends to show negative emotions [15], so for the M-type, when an individual’s negative mood score on the POMS questionnaire increases from morning to evening, its more positive personality traits and mental toughness may contribute to warding off the negative effects of changes in its circadian rhythm. However, the subjects of Vitale and other studies were college students, so whether the results are applicable to athletes or other people with mental illness remains to be explored. In addition, gender differences between studies may also lead to different results; compared with men, women are more likely to cope with frequent circadian rhythm disorders. Nevertheless, from the point of view that sleep time may indirectly regulate psychological reaction by affecting the physiological state of human organs and tissues, it is still suggested that night owls arrange courses in the afternoon or evening in future training to achieve the best emotional state.

### 5.3. The Influence of Emotional Circadian Rhythm on Sports Performance

In this review, several studies examined the effects of circadian variations in the mood on exercise performance. Two studies reported lower vigor before ME [29,32]. Essid et al. [29] additionally observed higher levels of negative emotions in the morning, and cognitive performance was also found to be greater in the AE/EE. Previous studies reported that individuals tend to be more reactive and alert in the evening, leading to mental cognitive performance peaking later in the day abilities [59,60]; it is generally believed that emotion status can affect an individual’s alertness, in addition to the perceptual choice of things, memory, and maintenance of stable attention, and through adjusting the speed of attention recognition and inhibition in cognitive processing, positive emotions can exert positive effects on executive control during physical activity [61]. Thus, it can be considered that the diurnal variation in mood is related to the circadian rhythm in cognitive performance during exercise, and positive emotions can positively regulate it. Grant et al. [31] found that higher anger and vitality in the morning did not produce positive benefits on handgrip strength and swimming performance among swimmers, suggesting that the diurnal variation in mood status was not responsible for different physical performance, which may be the result of circadian differences in skeletal muscle excitation–contraction coupling in the early morning and late evening in physiology [31]; Essid et al. reported similar results. Likewise, the differences in the dribbling sprint of ME/AE/EE were also not influenced by the circadian rhythm of emotion among children soccer players [33]. Therefore, it can be speculated that for athletes, the cognitive performance that requires more sensation, perception and thinking may be easily influenced by the day and night fluctuations of emotions, while physical abilities such as strength, endurance, and speed may not be. In order to achieve the best emotional state, it is suggested that physical function training or testing should also be synchronized with the circadian rhythm of individual emotions as much as possible.

There are several limitations to this review that should be noted. Firstly, there is considerable heterogeneity in the sample characteristics of the included studies, with the sample population covering both the general population without systematic training and high levels of professional athletes. Previous studies reported that the circadian rhythm amplitude of the former’s work performance is much lower than the latter [62]. In addition, few studies reported the nutritional status of subjects and whether subjects have circadian rhythm disorder events, which may affect the universality of the results. Furthermore, except for four experiments, most studies selected only the two-time points of morning and evening for comparison, as subjects might be exposed to varying degrees of sunlight and dietary factors during AE and EE, which may lead to a difference in the mood status response to physical activity at the two-time points; it is suggested that AE should be considered in future studies. Finally, much literature included in this study did not point out specific allocation hiding measures, which affected the quality of its methodology.

## 6. Conclusions

In this review, we systematically summarized the research on rhythmicity in sport psychology. There are many contradictions in the existing research, and it is speculated that the greater positive emotions the subjects experienced during or following ME may be due to their exposure to more sunlight, but given the greater muscle viscosity and lower body function in the morning, this may lead to higher levels of fatigue and lower vigor levels indirectly compared with EE. In addition, night owls seem to be more susceptible to the effects of exercise timing, and E-type individuals tend to show higher levels of negative emotions during morning training, which may be related to their personality traits and disturbed circadian rhythm. Furthermore, compared with cognitive performance, athletes’ physical fitness is more likely to be affected by the circadian rhythm of emotions. Therefore, in order to achieve the best performance, we suggest that athletes’ speed, strength, and other related sports training or testing should be synchronized with the circadian rhythm of individual emotions. In addition, in order to maximize the benefits of emotion in training, we suggest that night owls can arrange training in the afternoon, and the early birds seem to be free to choose their training time, but this still needs to be further verified by experiments. The sample size can be further increased in subsequent studies, and the measurement methods of outcome indicators can be standardized to improve the research design. Thus, it could provide the basis for coaches to optimize sports training scientifically and to improve the mental health of the related crowd to the greatest extent.

## Figures and Tables

**Figure 1 ijerph-20-02822-f001:**
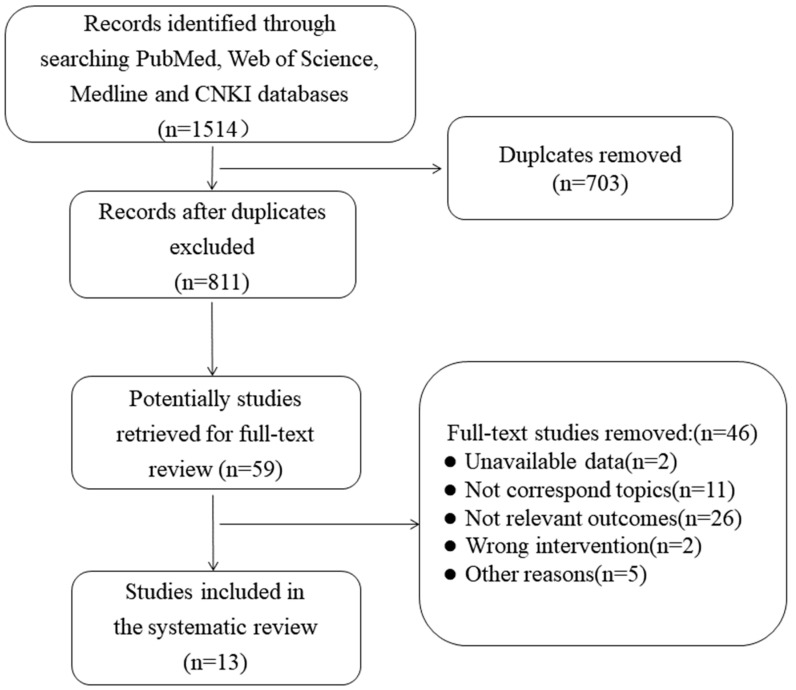
Flow chart of literature retrieval.

**Figure 2 ijerph-20-02822-f002:**
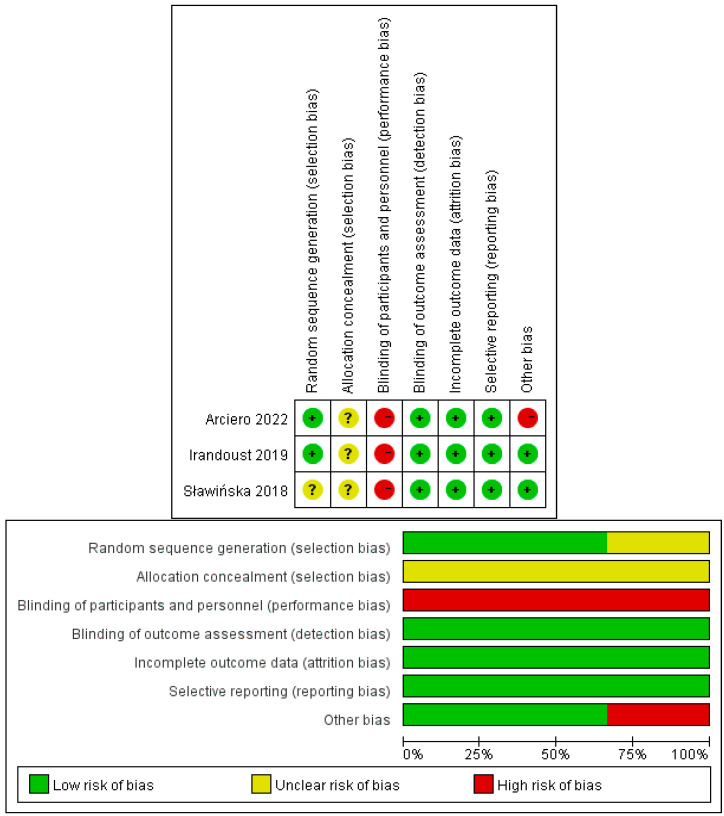
Risk of bias assessment for randomized controlled trials.

**Table 1 ijerph-20-02822-t001:** Risk of bias assessment for non-randomized controlled trials.

Author	Confounding Bias	Selection Bias	Bias in Measurement Classification of Interventions	Bias due to Deviations from Intended Interventions	Bias due to Missing Data	Bias in Measurement of Outcomes	Bias in Selection of the Reported Result
Essid 2021 [29]	moderate risk	low risk	low risk	low risk	low risk	low risk	low risk
Hill 2020 [30]	low risk	low risk	low risk	low risk	low risk	low risk	low risk
Grant 2018 [31]	moderate risk	low risk	low risk	low risk	low risk	moderate risk	low risk
Chtourou 2018 [32]	moderate risk	low risk	low risk	low risk	low risk	moderate risk	low risk
Vitale 2017 [9]	moderate risk	low risk	low risk	low risk	low risk	low risk	low risk
Masmoudi 2016 [33]	low risk	low risk	low risk	low risk	low risk	moderate risk	low risk
Rae 2015 [20]	low risk	low risk	low risk	low risk	low risk	moderate risk	low risk
Maraki 2005 [34]	moderate risk	low risk	low risk	low risk	low risk	low risk	low risk
Koltyn 1998 [35]	low risk	low risk	low risk	low risk	low risk	moderate risk	low risk
Mcmurray 1990 [36]	low risk	low risk	low risk	low risk	low risk	moderate risk	low risk

**Table 2 ijerph-20-02822-t002:** The relationship between chronobiology and sports psychology.

Study (Author, Year, Origin)	Subjects	Assessment of Chronotype	Exercise Timing	Intervention	Assessment of Mood Response	Main Results
Arciero et al. [37] 2022 USA	56 healthy adults (men: 45 ± 8 years; women: 42 ± 8 years)	not assessed	ME: 6:30–8:30 am EE: 6:00–8:00 pm	RISE 60 min/session 4 times/week 12 weeks 60%HR_max_	POMS	Women: No significant differences in mood status following intervention in the ME group from baseline (*p* < 0.05), and an interactive effect of mood disorders and tension was observed in the EE group (*p* > 0.05). Men: Anger, depression, tension, fatigue, confusion, and total mood disorders were significantly reduced following training (*p* < 0.05), and a more pronounced decrease in fatigue was observed in subjects in the EE group (*p* < 0.05).
Essid et al. [29] 2021 Tunisia	18 hanball players (16.5 ± 0.3 years)	MEQ	ME:10:00 am AE: 2:00 pm EE: 6:00 pm	RSA + BTV tests 30 min interval	POMS	The anxiety, confusion, depression, anger, and fatigue were more pronounced in the ME group (*p* < 0.001), while the EE group and the AE group showed greater levels of vigor status before tests (*p* < 0.001); the RSA and BTA were performed better in the afternoon and evening than in the morning (*p* < 0.001)
Hill et al. [30] 2020 USA	14 university students	MEQ	ME: 8:00 am AE: 2:00 pm EE: 6:00 pm	30s Wingate test	POMS	Anger, depression, fatigue, and total mood disturbance increased significantly in M-types during the day compared with E-types (*p* < 0.01), M-types showed a drop in vigor during the day while e-types showed a significant increase in it and a greater reduction in total mood disturbance (*p* < 0.01), the mean and peak power of Wingate test were significantly higher in the AE and the EE groups than in ME group(*p* < 0.01).
Irandoust et al. [38] 2021 Iran	71 retired male athletes (46.2 ± 2.1 years)	MEQ	ME: 8:00 am EE: 6:00 pm	Aerobic exercise(jogging) 80%HR_max_ 60 min/ session 3 times/ week 12 weeks	POMS	Compared with baseline, the depression, confusion, anger, tension, fatigue, and vigor were significantly improved in the subjects (*p* < 0.05), but there were no significant differences between the groups (*p* > 0.05).
Sławińska et al. [39] 2018 Poland	94 athletes (32 ± 6years)	CSM	ME: 6:30 am/7:45 am EE: 7:30 pm/8:45 pm	CrossFit training	EMAIL	The energetic arousal and hedonic tone improved and tension decreased significantly following exercise (*p* < 0.05), the morning training can raise the emotion of the E-type and N-type group to the level of M-type.
Grant et al. [31] 2018 UK	14 swimmers (14.8 ± 2.1 years)	MEQ	ME: 7:00–9:00 am EE: 6:00–8:00 pm	Muscle strength+ CMJ+ swimming performance test	POMS	No significant differences in swimming performance and jump height between the groups (*p* = 0.068, *p* = 0.756, respectively), the leg and back strength were significantly higher in the ME group (*p* = 0.013) while the hand grip strength was higher in the EE group (*p* = 0.007). The vigor and anger were significantly higher in the morning than in the evening (*p* = 0.000, *p* = 0.012, respectively).
Chtour-ou et al. [32] 2018 Tunisia	14 elite male judokas (21 ± 1 years)	MEQ	ME: 7:00 am EE: 5:00 pm	RSSJA	POMS	The vigor and jump height in CMJ3 and CMJ4 were significantly higher in the afternoon (*p* < 0.05), while the stress was greater in the morning, and no significant differences were found in other dimensions of mood (*p* > 0.05).
Vitale et al. [9] 2017 Italy	23 university students (21 ± 2years)	MEQ	ME: 8:00 am EE: 8:00 pm	HIIT 90~95% HR_max_	POMS	Compared to the M-type and the EE group, the E-type was more fatigued and showed less vigor during morning training (*p* < 0.05)
Masmo-udi et al. [33] 2015 Tunisa	10 male soccer players (14.6 ± 0.8 years)	MEQ	ME: 8:00 am AE: 1:00 pm EE: 5:00 pm	Shooting accuracy tests +10 × 20-m dribbling sprint	POMS	The RPE, depression and pressure scores were significantly greater in the evening (*p* < 0.05), there were no significant differences in anger, anxiety, vigor, fatigue, confusion, and the total mood score at different times of the day (*p* > 0.05). The dribbling performance of AE group and EE group was better than that of EE group (*p* < 0.05), but the differences in exercise performance between the groups cannot be explained by circadian variations in mood status.
Rae et al. [20] 2015 South Africa	26 swimmers (32.6 ± 5.7 years)	MEQ	ME: 6:30 am EE: 6:30 pm	200 m time trials	POMS	The ME group and M-type showed significantly lower fatigue and higher vigor before ME (*p* < 0.05),while the subjects accustomed to training at night and N-type showed lower fatigue before EE (*p* < 0.05), and the M-type worked better in ME (*p =* 0.036).
Maraki et al. [34] 2005 UK	12 healthy famales (18–45 years)	not assessed	ME:8:15–9:15 am EE:7:15–8:15 pm	Aerobic combined with muscle conditioning training 79% and 83.2% HR_max_	PANAS	The positive emotion in the ME group was significantly increased compared with the EE group and the morning control group (*p* = 0.005, *p* = 0.03, respectively), the ME group showed better effect of suppressing negative emotion than the EE group (*p* = 0.017).
Koltyn et al. [35] 1998 USA	16 men	not assessed	ME: 6:30–9:30 am EE: 5:00–8:00 pm	Constant power exhaustion training	POMS	The fatigue of the two groups increased significantly after exercise (*p* < 0.05), while the anger, anxiety, depression, tension, and comprehensive scores of emotion of the two groups did not change significantly (*p >* 0.05).
Mcmur- ray et al. [36] 1990 USA	14 healthy men (18–25 years)	MEQ	ME: 6:00 am 12:00 am EE: 6:00 pm 12:00 pm	Cycling 75VO%_2max_	GAS	The GAS score increased following training (*p* < 0.05), and the scores at 12:00 and 18:00 increased more significantly (*p* < 0.05).

ME: morning exercise; EE: evening exercise; RISE: resistance, interval, stretching, endurance; POMS: profile of mood states; MEQ: morningness–eveningness. questionnaire; AE: afternoon exercise; RSA: repeated sprint ability; BTV: ball-throwing velocity; HRmax: maximum heart rate (40% < low intensity < 60%, 60% < medium intensity < 70%, 70% < low intensity < 80%); CSM: composite scale of morningness; UMACL: UWIST mood adjective checklist; CMJ: countermovement jump; RSSJA: repeated shuttle sprint and jump ability; HIIT: high-intensity interval training, RPE: ratings of perceived exertion; PANAS: positive and negative affect schedule; VO2max:maximum oxygen uptake (low intensity < 40%, 40% < medium intensity < 60%, high intensity > 60%).

## Data Availability

No new data were created.

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
