# Peer review of "Mood Status Response to Physical Activity and Its Influence on Performance: Are Chronotype and Exercise Timing Affect?"

_ijerph, 2023, doi:10.3390/ijerph20042822_

Round 1

Reviewer 1 Report

Thank you for the opportunity to review the manuscript. The paper is interesting but needs some corrections, explanations, and additions. The main contribution of the authors is clearly presented. I have a few recommendations for the authors to address before publication.

My comments are as follows:

General comments:

1.  Authors in many places quote "Małgorzata". Please note that this is the author's first name and not her last name. In the case of citations, authors should use the surname of the author, in this matter "Sławińska".

2. The same issue occurs with other cited authors as well. This should be corrected. Authors should systematize this.

3. All references should be corrected according to MDPI Chicago Style. References are missing spaces, commas, etc. Please also remove "[J]." this information is inconsistent with MDPI guidelines. References should definitely be improved.

Abstract:

1. It is suggested to remove the name Małgorzata in the abstract. Perhaps this sentence could be reconstructed, as it confuses the reader.

Introduction

1. There are a lot of editing errors in the whole manuscript, e.g.

- Line 47: space is missing “nucleus(SCN)”

- Line 64 : space is missing “Questionnaire(MEQ)”; the opening bracket is missing “Questionnaire, MCTQ)”

- Line 94: space is missing “exercise(such)”

     Please check and correct the whole manuscript.

2.  Could authors specify what is new in this research, please?

3. What are the research questions? Maybe it would be good to post research questions to keep the article clearer to the reader.

Methods 

Literature Retrieval Strategy:

1. Why did the authors choose to include only literature from January 2002 to September 2022? Why were articles from before 2002 not included? I guess that needs some clarification.

2.  Could the authors clarify the following questions:

- Was a screening tool/form created for this project?

- Was pilot screen/calibration exercise done for Screeners?

- What was an agreement between the Screeners’ decisions on both stages of the screen?

3.  Full screen strategy for at least one database should be provided as an appendix (also with numbers) e.g. #search 1 – “elderly”, number of records.., #search 2 – “peer-led”, number of records…, #1 AND #2, number of records, etc.

4. Google Scholar provides a simple way to broadly search for scholarly literature. From one place, you can search across many disciplines and sources e.g. articles, theses, books, abstracts and court opinions, and other websites. Anyway, it’s not a database but rather a platform for searching in many databases – and as such is not appropriate methodologically. Therefore, there are some disadvantages while using this source, like problems with search precision and reproducibility.

Detailed information on this topic is available, for example, in the article:

- Gusenbauer M., Haddaway N.R. Which academic search systems are suitable for systematic reviews or meta-analyses? Evaluating retrieval qualities of Google Scholar, PubMed, and 26 other resources. Res Syn Meth. 2020;11:181–217. DOI: 10.1002/jrsm.1378

Perhaps authors should consider whether this is really the right search engine for this type of work. Maybe you should use more reliable search engines, eg BioMed Central, Medline (Pubmed), Embase (Ovid)?

5. Was the search string further expanded by running searches with Medical Subject Heading (MeSH Terms)? If so, it should be mentioned in the manuscript.

6. When the database search was performed? Could authors please specify this and include it in the manuscript?

Literature information extractions

1. Did you use a data extraction form based on the ICF framework during the data extraction process? Please complete this information in this section.

Results

Study Characteristics:

1. Lines 156-164: Please explain the meaning of all the abbreviations used, e.g. RSA, BTA, CMJ, and others.

2. Please systematize whether in Table 1 the authors of the articles are listed by surname or first name.

Outcomes

1. The outcomes would be much clearer if they were also presented in a graph or table. Perhaps it would be worthwhile to present the experiment graphically for better understanding. If there are practical results to be highlighted, it may make the results more clear.

Discussion

1. The Discussion should also provide practical guidelines for the reader.

Conclusions

1. While the practical applications are almost obvious, perhaps they should be mentioned.

2. The inability to reach a consensus on the results of the research is a definite disadvantage of this work. Perhaps the scope of the search should be extended to include the work of other authors. There is a possibility that this would shed a different light on the results and conclusions.

Reference

1. Please correct the references according to MDPI Chicago Style.

Author Response

Dear experts:

First of all, I apologize to you, because I was infected with COVID-19, and my body was very weak, so the article did not return within 10 days (the reason has been explained to the editor). Thank you very much for your valuable comments on the revision of this article. This revision has been carried out in strict accordance with your comments one by one. We have marked the revised places in red font.

Thank you again and best wishes!

General comments:

  1. Authors in many places quote "Małgorzata". Please note that this is the author's first name and not her last name. In the case of citations, authors should use the surname of the author, in this matter "Sławińska".

According to your requirements, we have modified and replaced Małgorzata with Sławińska.

  1. The same issue occurs with other cited authors as well. This should be corrected. Authors should systematize this.

Thank you for pointing out our mistakes. According to your requirements, we have modified them. We replaced Małgorzata with Sławińska;We replaced Sana with Essid;We replaced David with Hill;We replaced Khadijah with Irandoust;We replaced Marie with Grant;We replaced Hamdi with Chtourou;We replaced Liwa with Masmoudi.

  1. All references should be corrected according to MDPI Chicago Style. References are missing spaces, commas, etc. Please also remove "[J]." this information is inconsistent with MDPI guidelines. References should definitely be improved.

Thank you for pointing out our mistakes. According to your requirements, we have modified them.

Abstract:

  1. It is suggested to remove the name Małgorzata in the abstract. Perhaps this sentence could be reconstructed, as it confuses the reader.

According to your request, we deleted Małgorzata in the abstract. We changed it to:while one recorded with the CSM.

Introduction

  1. There are a lot of editing errors in the whole manuscript, e.g.

- Line 47: space is missing “nucleus(SCN)”

According to your request, we added the space.

- Line 64 : space is missing “Questionnaire(MEQ)”; the opening bracket is missing “Questionnaire, MCTQ)”

According to your request, we added the space and bracket.

- Line 94: space is missing “exercise(such)”

According to your request, we added the space.

Please check and correct the whole manuscript.

Thank you for your correction. We have carefully checked the full text and revised the inappropriate symbols

  1. Could authors specify what is new in this research, please?

4.What are the research questions? Maybe it would be good to post research questions to keep the article clearer to the reader.

According to your request, we pointed out the main problem to be solved in this article in the third paragraph of the introduction. The research on the rhythmicity of the exercise system has been studied for several years, but it mainly focuses on the exercise physiology, that is, how the exercise time of the day affects the exercise performance and metabolic results. However, there are few reviews involving the field of exercise psychology. Based on this, We pointed out the main problems of this paper:However, it remains unclear how the time of day and chronotype regulates mood states' response to physical activity and how the circadian rhythm of mood affects performance. Based on this, the systematic review summarized the evidence on rhythmicity in sports psychology.

Methods 

Literature Retrieval Strategy:

  1. Why did the authors choose to include only literature from January 2002 to September 2022? Why were articles from before 2002 not included? I guess that needs some clarification.

According to your request, we re-retrieved the database. This time, we included all the studies before 2022, and finally obtained 13 articles (two more than before, and we included them in this review).

  1. Could the authors clarify the following questions:

- Was a screening tool/form created for this project?

According to your request, We pointed out the screening tool in 2.3:“Using Endnote x9 for document screening and management”

- Was pilot screen/calibration exercise done for Screeners?

We conducted prior practice for the screening staff, and we declare in 2.3: two reviewers (LJQ and LSQ, they performed screening exercises in advance) independently screened the literature.

- What was an agreement between the Screeners’ decisions on both stages of the screen?

According to your request, we declare in 2.3: After the first screening, the consistency among screeners reached 92.3%, and came to 100% after the third party's discussion on the dispute.

  1. Full screen strategy for at least one database should be provided as an appendix (also with numbers) e.g. #search 1 – “elderly”, number of records.., #search 2 – “peer-led”, number of records…, #1 AND #2, number of records, etc.

#1 Mood state OR Psychological response OR Mental state OR Emotional state

#2 Physical activity OR Exercise OR Training OR Sports OR Sports performance OR Exercise performance

#3 Chronotype OR Exercise timing OR Time of day OR Circadian rhythm OR Diurnal variation OR Diurnal rhythms.

#4 #1 AND#2 AND#3

In addition, we screened the reference lists from the included literature for potential studies that had not been previously retrieved.

  1. Google Scholar provides a simple way to broadly search for scholarly literature. From one place, you can search across many disciplines and sources e.g. articles, theses, books, abstracts and court opinions, and other websites. Anyway, it’s not a database but rather a platform for searching in many databases – and as such is not appropriate methodologically. Therefore, there are some disadvantages while using this source, like problems with search precision and reproducibility.

Detailed information on this topic is available, for example, in the article:

- Gusenbauer M., Haddaway N.R. Which academic search systems are suitable for systematic reviews or meta-analyses? Evaluating retrieval qualities of Google Scholar, PubMed, and 26 other resources. Res Syn Meth. 2020;11:181–217. DOI: 10.1002/jrsm.1378

Perhaps authors should consider whether this is really the right search engine for this type of work. Maybe you should use more reliable search engines, eg BioMed Central, Medline (Pubmed), Embase (Ovid)?

Thank you very much for pointing out our mistakes. We carefully read the references you gave us and made modifications according to your requirements. We replaced "Google Scholar" with "Medline", and then re-researched the database, We made statements in the abstract and 3.2, taking 3.2 as an example:We searched pub med, Web of Science, Medline and CNKI databases for relevant literature ,the search scope is research before September 2022. Results: 13 studies comprising 382 subjects examined the effects of exercise timing on mood responses to exercise, or the effects of circadian rhythms of mood on exercise performance, which included 3 RCTs and 10 Non-RCTs. The subjects included athletes (both training or retired), college students, and healthy adults. Two studies were designed for long-term exercise intervention (aerobic training and RISE), and the rest for acute intervention (CrossFit training, HIIT, constant power exhaustion training and cycling)or physical function tests (RSA+BTV tests, 30s Wingate test, muscle strength+CMJ+swimming performance test, RSSJA, shooting accuracy tests+10×20m dribbling sprint, 200m time trials). All trials reported specific exercise timing, of these,10 studies reported subjects’ chronotypes, most commonly using the MEQ scale, while one recorded with the CSM. Mood responses were assessed with the POMS scale in 10 studies, while three other studies used UMACL, PANAS and GAS scales, respectively.

  1. Was the search string further expanded by running searches with Medical Subject Heading (MeSH Terms)? If so, it should be mentioned in the manuscript.

According to your request, we made a statement in the text: the retrieval strategy is based on the combination of subject words and free words, and using boolean operators "AND" and "OR" to connect.

  1. When the database search was performed? Could authors please specify this and include it in the manuscript?

According to your request, We made a statement in 2.1:Two reviewers (LHX and LJQ) searched Pub med, Web of Science, Medline, and CNKI databases for relevant literature on November 2, 2022.

Literature information extractions

  1. Did you use a data extraction form based on the ICF framework during the data extraction process? Please complete this information in this section.

According to your request, we have made a statement in the text: they performed screening exercises in advance) independently screened the literature, extracted data (using a data extraction form based on the ICF framework) and cross-checked it.

Results

Study Characteristics:

  1. Lines 156-164: Please explain the meaning of all the abbreviations used, e.g. RSA, BTA, CMJ, and others.

According to your requirements, we marked these abbreviations in 3.2: Two studies were designed for long-term exercise intervention (aerobic exercise and resistance, interval, stretching, endurance (RISE) training), and the rest for acute in-tervention (CrossFit training, high intensity interval training (HIIT), constant power exhaustion training and cycling)or physical function tests (repeated sprint ability (RSA) + ball-throwing velocity (BTV) tests, 30s Wingate test, muscle strength+countermovement jump (CMJ) +swimming performance test, repeated shuttle sprint and jump ability (RSSJA), shooting accuracy tests+10×20m dribbling sprint and 200m time trials). All trials reported specific exercise timing, of these, 9 studies reported subjects’ chronotypes, most commonly using the MEQ scale, while Sławińska recorded with the composite scale of morningness (CSM). Mood responses were assessed with the profile of mood states (POMS) scale in 9 studies, while two other studies used UWIST mood adjective checklist (UMACL) and positive and negative affect schedule (PANAS) scales, respectively.

  1. Please systematize whether in Table 1 the authors of the articles are listed by surname or first name.

According to your requirements, we have modified them: We replaced Małgorzata with Sławińska;We replaced Sana with Essid;We replaced David with Hill;We replaced Khadijah with Irandoust;We replaced Marie with Grant;We replaced Hamdi with Chtourou;We replaced Liwa with Masmoudi.

Outcomes

  1. The outcomes would be much clearer if they were also presented in a graph or table. Perhaps it would be worthwhile to present the experiment graphically for better understanding. If there are practical results to be highlighted, it may make the results more clear.

Thank you for your question, but I'm very sorry that I didn't understand your meaning because of my limited understanding. Could you explain it in more detail so that we can continue to revise it. Because we mainly describe the contents of the table (i.e. the documents we included) in the results section, our table refers to several documents previously published on MDPI, and displays the included documents according to their format.

Discussion

  1. The Discussion should also provide practical guidelines for the reader.

According to your request, we have added some suggestions in the discussion section (at the end of 5.1, 5.2 and 5.3):

5.1: which suggested that the response of emotional state to exercise training may be tran-sient, and the influence of circadian rhythm will gradually disappear when the training period is extended. In addition, it is worth noting that the subjects in Irandoust and other studies are depressed people, due to the fact that these patients have obvious characteristics of severe morning symptoms and mild night onset, so there is also a possibility that the improvement of ME on depression may be covered up by its own circadian rhythm rise, which may also lead to insignificant differences between groups. Furthermore, the subjects’ level of training, sex (as observed in the study by Arciero et al[29]), and age as well as the heterogeneity of exercise intervention type and intensity may all lead to different results.

5.3: Therefore, it can be speculated that for athletes,the cognitive performance that requires more sensation, perception and thinking may be easily influenced by the day and night fluctuations of emotions, while physical abilities such as strength, endurance and speed may be not. In order to achieve the best emotional state, it is suggested that physical function training or testing should also be synchronized with the circadian rhythm of individual emotions as much as possible.

Conclusions

  1. While the practical applications are almost obvious, perhaps they should be mentioned.

According to your request, we have added practical applications in the conclusion. Furthermore, compared with cognitive performance, athletes' physical fitness is more likely to be affected by the circadian rhythm of emotions. Therefore, in order to achieve the best performance, we suggest that athletes' speed, strength and other related sports training or testing should be synchronized with the circadian rhythm of individual emotions. In addition, in order to maximize the benefits of emotion in training, we suggest that night owls can arrange training in the afternoon, and the early birds seem to be free to choose their training time, but this still needs to be further verified by experiments.

  1. The inability to reach a consensus on the results of the research is a definite disadvantage of this work. Perhaps the scope of the search should be extended to include the work of other authors. There is a possibility that this would shed a different light on the results and conclusions.

According to your request, we have expanded the search scope, and the number of included documents has increased from 11 to 13. Due to the large heterogeneity of the subjects' age, gender, whether they are athletes, and the intensity, cycle and type of sports intervention, we also try to explain the heterogeneity before reaching some conclusions. For example:

5.2: However, the subjects of Vitale and other studies are college students, so whether the results are applicable to athletes or other people with mental illness remains to be explored. In addition, gender differences between studies may also lead to different results, compared with men, women are more likely to cope with frequent circadian rhythm disorders.

Reference

  1. Please correct the references according to MDPI Chicago Style.

According to your requirements, we have modified them.

Thank you again for your valuable comments and best wishes!

Reviewer 2 Report

The review has a very interesting theme and was meant to bring clarity to the field of physical activity regarding the time frames used for developing this process. Unfortunately, in my opinion, the result is at least neutral and unsatisfactory, as the studies taken into consideration for further assessment are rather diverse, opposite concerning the typology of subjects included in their groups of study, and bringing together 11 very different trials brings no homogeneity in the results, and incoherent discussions and conclusions.

I could not depict any decent classification regarding the type of effort implied by these subjects, of the particular sports, practised by them, whether it was a long-term type of effort or an acute, short-term one, high vs. low-intensity programmes.

Another major limitation and drawback of the review is that no other environmental and metabolic factors were considered throughout the interpretations: nutrition, sex, age, previous circadian disruptive events, previous history regarding physical activities, etc. 

I claim the inconsistency, flagrant heterogeneity and chaotic disposal of data, and I advise the authors to rewrite the review in a more restrained but organised manner to conduct correct and pertinent conclusions. otherwise, the scientific usefulness is completely lacking despite the interesting and important subject.

I also suggest the authors to show their personal input in elaborating the conclusions, to make a clear statement and boldly bring to the surface the outcomes of their work.

Author Response

Dear experts:

First of all, I apologize to you, because I was infected with COVID-19, and my body was very weak, so the article did not return within 10 days (the reason has been explained to the editor). Thank you very much for your valuable comments on the revision of this article. This revision has been carried out in strict accordance with your comments one by one. We have marked the revised places in red font.

Thank you again and best wishes!

The review has a very interesting theme and was meant to bring clarity to the field of physical activity regarding the time frames used for developing this process. Unfortunately, in my opinion, the result is at least neutral and unsatisfactory, as the studies taken into consideration for further assessment are rather diverse, opposite concerning the typology of subjects included in their groups of study, and bringing together 11 very different trials brings no homogeneity in the results, and incoherent discussions and conclusions.

we expanded the search scope, we also include the literature before 2002 (formerly 2002-2022) ,and the number of included documents has increased from 11 to 13. Due to the large heterogeneity of the subjects' age, gender, whether they are athletes, and the intensity, cycle and type of sports intervention, we also try to explain the heterogeneity before reaching some conclusions according to your requirements. We rewrote the discussion section and divided it into three parts:5.1 Time dependence influence of training on emotional state 5.2 The effect of individual chronotype on psychological reaction to exercise 5.3 The influence of emotional circadian rhythm on sports performance. In addition, because each part is included in the limited order of magnitude of the literature, the comparability between the documents is poor. Therefore, at the end of each part, we are not only giving conclusions and suggestions, but also cautioning readers to carefully explain the research results because of the large heterogeneity between the studies

I could not depict any decent classification regarding the type of effort implied by these subjects, of the particular sports, practised by them, whether it was a long-term type of effort or an acute, short-term one, high vs. low-intensity programmes.

According to your request, we have reclassified and explained the sports types and intensity included in the literature. We divided the literature into long-term regular exercise intervention, acute exercise and physical function test. First of all, according to the intervention cycle, the intervention cycle of Arciero and Irondoust and other studies is 12 weeks, so they are long-term exercise intervention. Among the other studies, 6 are physical function tests (RSA+BTV tests, 30s Wingate test, muscle strength+CMJ+swimming per-formance test, RSSJA, shooting accuracy tests+10 × 20 m dribbling sprint, 200 m time trials), and the rest are acute exercise (single) intervention. Among them, three studies measured intensity related indicators, such as the maximum oxygen uptake and the maximum heart rate. We have marked them in the table, and we have declared the intensity classification criteria in the note: HRmax: maximum heart rate (40% < low intensity < 60%, 60% < medium intensity < 70%, 70% < low intensity < 80%); VO2max:maximum oxygen uptake (low intensity<40%, 40%<medium intensity<60%, high intensity>60%).

Another major limitation and drawback of the review is that no other environmental and metabolic factors were considered throughout the interpretations: nutrition, sex, age, previous circadian disruptive events, previous history regarding physical activities, etc. 

According to your requirements, we have made a statement on these limitations in the article, Since few studies have considered the nutrition of subjects and whether there are circadian rhythm disorders, we mentioned in the limitations of this article: In addition, few studies reported the nutritional status of subjects and whether subjects have circadian rhythm disorder events, which may affect the universality of the results.

Furthermore, in each part of 5.1, 5.2 and 5.3, when comparing the literature, we explained the variables that may affect the results

5.1:In addition, it is worth noting that the subjects in Irandoust and other studies are de-pressed people, due to the fact that these patients have obvious characteristics of severe morning symptoms and mild night onset, so there is also a possibility that the im-provement of ME on depression may be covered up by its own circadian rhythm rise, which may also lead to insignificant differences between groups. Furthermore, the subjects’ level of training, sex (as observed in the study by Arciero et al[29]), and age as well as the heterogeneity of exercise intervention type and intensity may all lead to different results.

5.2:However, the subjects of Vitale and other studies are college students, so whether the results are applicable to athletes or other people with mental illness remains to be ex-plored. In addition, gender differences between studies may also lead to different results, compared with men, women are more likely to cope with frequent circadian rhythm disorders.

5.3:Firstly, there is considerable heterogeneity in the sample characteristics of the included studies, with the sample population covering the both general population without systematic training and high levels of professional athletes, previous studies reported that the circadian rhythm amplitude of the former work performance is much lower than the latter[66]

I claim the inconsistency, flagrant heterogeneity and chaotic disposal of data, and I advise the authors to rewrite the review in a more restrained but organised manner to conduct correct and pertinent conclusions. otherwise, the scientific usefulness is completely lacking despite the interesting and important subject.

In the study included in Part 5.1, there are differences mainly in the type of subjects (one study is a depressive population), cycle (one is a long-term study) and gender. Therefore, we also discussed the limitations caused by these variables when we put forward the conclusion at the end of Part 5.1.

In the study included in section 5.2, there are differences mainly in the type of subjects and gender, so we also discussed the limitations when we put forward the conclusion at the end of section 5.2

Since the subjects of the study included in 5.3 are athletes, and the results tend to be similar, our conclusions in this part are mainly aimed at athletes, that is, they can be specified that for athletes, the corresponding performance that requires more sensing, perception and thinking may be easily influenced by the exercise timing, while physical abilities such as strength, endurance and speed may be not. In order to achieve the best emotional state, it is suggested that physical function training or testing should also be synchronized with the circadian rhythm of individual emotions as much as possible.

I also suggest the authors to show their personal input in elaborating the conclusions, to make a clear statement and boldly bring to the surface the outcomes of their work.

According to your request, we showed our conclusions and suggestions in the text:

5.1: Thus, it can be seen that there is still some controversy about whether the impact of acute exercise on emotional state is time-dependent.

which suggested that the response of emotional state to exercise training may be transient, and the influence of circadian rhythm will gradually disappear when the training period is extended.

5.3:Therefore, it can be speculated that for athletes,the cognitive performance that requires more sensation, perception and thinking may be easily influenced by the day and night fluctuations of emotions, while physical abilities such as strength, endurance and speed may be not. In order to achieve the best emotional state, it is suggested that physical function training or testing should also be synchronized with the circadian rhythm of individual emotions as much as possible.

Conclusions: Furthermore, compared with cognitive performance, athletes' physical fitness is more likely to be affected by the circadian rhythm of emotions. Therefore, in order to achieve the best performance, we suggest that athletes' speed, strength and other related sports training or testing should be synchronized with the circadian rhythm of individual emotions. In addition, in order to maximize the benefits of emotion in training, we suggest that night owls can arrange training in the afternoon, and the early birds seem to be free to choose their training time, but this still needs to be further verified by experiments.

Thank you again and best wishes!

Round 2

Reviewer 2 Report

The comments are valuable and the changes provided to the article improved it considerably.